# Hierarchical Principal Components for Data-Driven Multiresolution fMRI Analyses

**DOI:** 10.3390/brainsci14040325

**Published:** 2024-03-28

**Authors:** Korey P. Wylie, Thao Vu, Kristina T. Legget, Jason R. Tregellas

**Affiliations:** 1Department of Psychiatry, School of Medicine, University of Colorado Anschutz Medical Campus, Aurora, CO 80045, USA; kristina.legget@cuanschutz.edu (K.T.L.); jason.tregellas@cuanschutz.edu (J.R.T.); 2Department of Biostatistics and Informatics, Colorado School of Public Health, University of Colorado Anschutz Medical Campus, Aurora, CO 80045, USA; 3Research Service, Rocky Mountain Regional VA Medical Center, Aurora, CO 80045, USA

**Keywords:** treelets, hPCA, independent component analysis, simulation, multiscale, hierarchy, functional connectivity

## Abstract

Understanding the organization of neural processing is a fundamental goal of neuroscience. Recent work suggests that these systems are organized as a multiscale hierarchy, with increasingly specialized subsystems nested inside general processing systems. Current neuroimaging methods, such as independent component analysis (ICA), cannot fully capture this hierarchy since they are limited to a single spatial scale. In this manuscript, we introduce multiresolution hierarchical principal components analysis (hPCA) and compare it to ICA using simulated fMRI datasets. Furthermore, we describe a parametric statistical filtering method developed to focus analyses on biologically relevant features. Lastly, we apply hPCA to the Human Connectome Project (HCP) to demonstrate its ability to estimate a hierarchy from real fMRI data. hPCA accurately estimated spatial maps and time series from networks with diverse hierarchical structures. Simulated hierarchies varied in the degree of branching, such as two-way or three-way subdivisions, and the total number of levels, with varying equal or unequal subdivision sizes at each branch. In each case, as well as in the HCP, hPCA was able to reconstruct a known hierarchy of networks. Our results suggest that hPCA can facilitate more detailed and comprehensive analyses of the brain’s network of networks and the multiscale regional specializations underlying neural processing and cognition.

## 1. Introduction

One of the major challenges in neuroscience is understanding the organization and function of the brain’s neural processing systems. A defining feature of this organization is spatial heterogeneity. In this framework, localized specialized regions are encompassed within larger processing systems. The resulting topology is organized as a spatial hierarchy, with a decreasing spatial scale corresponding to increasingly specialized processing [1,2,3,4]. In detailing this hierarchy below, the term “region” will denote a circumscribed and spatially contiguous cortical area, “subnetwork” will denote a subset of regions within a larger distributed network, and “processing system” will encompass both terms as being situated within a larger network with a known neurocognitive function.

As an example of this hierarchy, the visual processing system encompasses the occipital lobe and adjacent regions of the temporal and parietal lobes. Within the larger visual system, the primary visual cortices along the calcarine sulcus specialize in processing low-level sensory information from the thalamus, while dorsal and ventral streams specialize in higher-level processing involving movement detection and object recognition, respectively [5]. Within the primary, dorsal, and ventral visual subsystems are even smaller cortical regions, each with a more specialized function [6,7]. Consequentially, the visual system is organized as a spatial hierarchy. The largest spatial scale is the occipital lobe as a whole and its role in processing general visual information, with nested smaller subsystems processing specialized aspects of visual information.

Similar nested spatial hierarchies have been observed in the somatosensorimotor processing system. In this case, within the larger primary sensorimotor cortices are regions dedicated to specific parts of the body, such as the legs, hands, and face. These more specialized processing regions form the sensory and motor homunculi of the pre-central and post-central gyri [5]. Nested within each are even more specialized processing regions, such as cortical representations of the individual fingers of the hand or parts of the face. Recent work has refined the motor homunculi with the addition of regions specialized for intercostal muscles [8] and whole-body movement planning [9], as well as highlighting potential overlap between the somatotopic representations of the parts of the body within the primary motor cortex [10]. In addition to visual and somatosensory systems, the auditory processing system features a hierarchical organization as well [11].

Beyond primary sensory systems, functional connectivity analyses using functional magnetic resonance imaging (fMRI) have found that spatial hierarchies appear to extend to associative processing systems such as the Default Mode Network (DMN) and Dorsal Attention Network (DAN) [12]. Each subnetwork encompasses circumscribed cortical regions, although the associated specialized processing roles for these subnetworks are less well established than in the sensory subnetworks and regions. This organization suggests that spatial hierarchies may represent a general organizing principle for specialized processing within the brain [13], although it is by no means unique in this role [14,15,16,17].

Current functional connectivity methods typically are limited to investigating a single scale of the brain’s multiscale hierarchy. Independent component analysis (ICA), for example, is frequently used to investigate large-scale networks [18,19]. The spatial scale is dependent, however, on the expected number of networks chosen as an a priori parameter [20]. The most commonly used choice, 20 networks [21,22], successfully estimates large-scale networks for major processing systems, such as the Visual and Sensorimotor Networks. However, relatively smaller networks, such as the dorsal and ventral visual streams or individual regions of the sensorimotor homunculi, are not estimated in these analyses [23]. In this case, the spatial overlap with the larger networks results in large statistical dependencies and thus violates the independence required for estimation with ICA. Similarly, the individual occipital and parietal regions encompassed by these subnetworks can be investigated using higher-model-order ICA [20,24]. These more fine-grained results do not, however, include the larger Visual and Sensorimotor Networks due to spatial overlap and the resulting statistical dependency.

In addition to ICA, applying a region-of-interest atlas or network parcellation atlas also facilitates investigation at a single spatial scale [2]. Many current neuroimaging investigations apply such atlases, with results that advance our understanding of neural processing at a single scale. Indeed, the conceptual clarity and computational simplicity provided by parcellation atlases are considerable. However, this approach is not without limitations. Processing at smaller scales will likely not be identified due to regional averaging, while processing at larger scales can be overlooked due to the narrow analytic focus. Furthermore, recent investigations at the level of individual subjects have suggested that functional mapping features greater regional variability than has been previously appreciated, raising doubts on the assumption of a shared group-level region of interest [25]. As such, new analytic methods are needed to investigate the full range of organization within the brain and its spatial processing hierarchies.

One method to examine the brain’s organization across multiple scales is hierarchical principal component analysis (hPCA) [26], which has been recently introduced as a data-driven multiresolution analysis technique but has yet to be applied to fMRI data. Hierarchical PCA extends multiresolution wavelet analysis to unordered data, such as information from connectivity analyses in neuroimaging. At each of the multiple levels at which data are examined, hPCA constructs increasingly generalized features of the data by merging smaller networks or other variables using a local PCA (Figure 1). Lower levels estimate localized data features, such as a time series averaged over all voxels within a region of interest. Higher levels estimate more global features of the data, such as time series resulting from activity within an entire large-scale network. Notably, although global PCA continues to be extensively used as a data compression method in fMRI, the nested local PCAs of hPCA make it a distinct method from the single global PCA that is frequently applied in neuroimaging.

Hierarchical PCA is similar to an agglomerative hierarchical clustering algorithm in many respects [27]. However, the two methods differ in important details, including the potential overlap between clusters and latent variables. Notably, rather than grouping together input variables as in clustering algorithms, hPCA systematically constructs new latent variables that summarize increasingly generalized features of the data. When hierarchical clustering analysis is applied to spatial fMRI, each voxel is located within one and only one cluster, termed “hard clustering”. Hierarchical clustering algorithms thereby rigidly group voxels into clusters, without investigating any potential underlying variables that may generate such patterns. In contrast, at each level, hPCA generates a latent variable that contributes to multiple voxels as a network time series. As a consequence, any individual voxel can feature a linear combination of such time series and any network time series can influence multiple voxels. Additionally, hPCA simultaneously constructs both a dendrogram and set of basis functions, whereas hierarchical clustering algorithms only construct the former. Lastly, the basis functions from hPCA facilitate orthogonalization of the time series, a feature capitalized on in the current work. Hierarchical PCA may thus overcome the limitations of previous functional connectivity methods, potentially allowing investigations into the full spatial processing hierarchy observed with functional connectivity in fMRI.

The current study aimed to investigate the accuracy of applying hPCA to fMRI data. Since the network structure of the brain is only partially understood, accuracy was assessed using simulated fMRI datasets with a known hierarchical structure as the baseline “ground truth” in comparisons. Following the validation procedure developed for ICA [18,19], accuracy was assessed in both the spatial and temporal domains of individual subjects (i.e., each subject’s unique spatial maps and time series, respectively). Concurrently, a method of combining subjects for group analyses while allowing for substantial individual variation was developed for hPCA. Additionally, hPCA was applied to data from the Human Connectome Project (HCP), with a largely unknown hierarchical structure, in order to validate its performance on a real fMRI dataset. Network spatial maps and time series were qualitatively and quantitatively compared with ICA, describing each analytic method’s strengths and weaknesses. Lastly, statistical methods for hPCA were developed to discard redundant information and artifactual noise inadvertently incorporated into the hierarchy, focusing the analyses on neurobiologically relevant features. We hypothesized that, compared to ICA, hPCA would more accurately estimate spatial maps and time series.

## 2. Materials and Methods

In the following paper, Section 2.1 describes the procedure used to create synthetic fMRI datasets. This section includes descriptions of the three hierarchies of networks modeled, along with the reasoning supporting each comparison. Section 2.2 then details how subjects from these datasets were combined to facilitate group-level analyses, including for ICA and hPCA. Section 2.3 provides an overview of the ICA method applied. Section 2.4 describes the hPCA algorithm in detail. Section 2.5 details the novel statistical framework developed for hPCA, in order to select a relevant subset of levels for focused analyses from a potentially very large hierarchy. Section 2.6 describes how individual subject-level results for these levels were back-reconstructed from the group-level results. Section 2.7 details the methods used to assess and compare the accuracy of the ICA and hPCA results. Section 2.8 provides an overview of the real fMRI dataset used to validate hPCA via comparison with known large-scale networks. Section 2.9 provides the code used in analyses, including that used to generate the simulated datasets and the hPCA algorithm.

### 2.1. Simulated Datasets and Hierarchical Structure

Data for simulated fMRI subjects were generated using the fMRI simulation toolbox SimTB v18, https://trendscenter.org/software/ (accessed on 26 March 2024) [28]. SimTB subject data are a linear mixture of source components scaled by amplitude,
(1)$$Z^{nf}=SDR,$$
where **Z**^nf^ is the voxel-by-time noise-free (nf) data (**Z**) for a single subject, **S** is the matrix of spatial maps in columns, **R** is the matrix of component time series in rows, and **D** is a diagonal matrix of component amplitudes, defined as percent signal change from the baseline. As detailed below, the total number of components varied in each experimental dataset depending on the depth and branching in each simulated hierarchy. In this model, subject-specific spatial maps in **S** are generated from shared group spatial maps with small amounts of added noise drawn from a normal distribution N(0, 0.005). Consistent with the blood oxygen level-dependent (BOLD) model of fMRI, component time series in **R** are generated from underlying neural events drawn from Bernoulli distribution B(0.2) at each time point and filtered through a hemodynamic response function developed for fMRI [29]. The resulting subject-specific time series are then normalized to a mean of 0 and a peak-to-peak range of 1. For each subject, a contrast-to-noise ratio (CNR) is drawn from a uniform distribution U(0.65, 2), a temporal standard deviation *σ_s_* of the true signal is calculated from the data, and a noise standard deviation is calculated as *σ_n_* = *σ_s_*/CNR. Lastly, to simulate fMRI scanner conditions, Rician noise [30] is added to each time point *t* for voxel *v*,
(2)$$Z_{vt}=\sqrt{{\left(Z_{vt}^{nf}+\varepsilon_{1}\right)}^{2}+\varepsilon_{2}^{2}},$$
where *ε*_1_ and *ε*_2_ are drawn independently from N(0, *σ_n_*^2^) for each voxel and time point. The resulting matrix **Z** = [*Z_vt_*] is the voxel-by-time matrix with added noise for an individual simulated subject. In all datasets, networks with different hierarchical structures were simulated by varying the group spatial maps used to generate **S**.

Simulated data were created for three experiments, each with 100 subjects, 1024 voxels, 300 time points, and a repetition time (TR) of 2 s. To simulate component heterogeneity, source amplitudes were distributed as N(1, 0.3), consistent with sources in real fMRI data [28]. Default SimTB settings were used for all other parameters.

Three experiments were chosen with hierarchies presenting potential challenges to hPCA to accurately model the potentially diverse branching structure in real fMRI data. Hierarchical structures were varied using three parameters: (1) the total number of levels (denoted by L in all experiment names), (2) the degree of branching at each level (denoted by D), and (3) the size of the blocks at each level, in terms of the relative distribution of voxels at each branch. Block sizes were either equally subdivided at each branching point (e.g., three-degree branching would divide all branches by 1/3 at each level) or randomly drawn from a Dirichlet distribution (dirich) at each branch and level. Experiments were systematically named based on these parameters. One experiment featured a relatively deep hierarchy (experiment 5L2D), a second experiment featured a hierarchy with an odd branching structure (experiment 3L3D), and a third experiment featured widely varying voxel numbers in each branch subdivision (experiment 3L3Ddirich). For each experiment, group spatial maps with a nested hierarchical structure were created using custom scripts, https://github.com/koreywylie/SimTB_hierarchies (accessed on 26 March 2024). The resulting group-averaged correlation matrices for all three experiments are displayed in Figure 2.

Experiment 5L2D simulated a spatial hierarchy with five levels, two-way branching at each level, and equal block sizes. This experiment was designed to test the ability of hPCA to reconstruct hierarchies with a simple branching structure but greater depth than other experiments. Despite the greater depth, this experiment is arguably the simplest and most straightforward due to the balanced binary subdivisions at each level. There were 62 components in total in this experiment (i.e., two-way branching over five levels results in 2^1^ + 2^2^ + 2^3^ + 2^4^ + 2^5^ = 62 total components).

Experiment 3L3D simulated a spatial hierarchy with three levels, three-way branching at each level, and equal block sizes. Since the hPCA algorithm reconstructs a spatial hierarchy with a two-way merger at each level, experiment 3L3D was designed to test the ability of hPCA to reconstruct hierarchies with an odd degree of branching at each level. In this experiment, hPCA was predicted to reconstruct the true three-way branching at each level with paired two-way mergers across two hPCA levels. There were 39 components in total in this experiment (3^1^ + 3^2^ + 3^3^).

Experiment 3L3Ddirich simulated a spatial hierarchy with three levels, three-way branching at each level, and randomly distributed block sizes drawn from a Dirichlet distribution with parameter alpha = (3,3,3). This more complicated experiment was designed to test the ability of hPCA to reconstruct hierarchies with uneven block sizes, modeling the spatial heterogeneity observed in fMRI. There were 39 components in total in this experiment (3^1^ + 3^2^ + 3^3^).

### 2.2. Data Compression and Multi-Subject Methods

Prior to both the ICA and hPCA, the size of the dataset was reduced and subjects were combined using a procedure developed for ICA [31]. This two-step procedure is designed to facilitate analyses of shared features at the group level, while allowing such features to vary widely at the level of the individual subject.

Let **Z***_i_* be the *V*-by-*N* uncompressed data matrix for subject *i*, with *V* voxels arranged in rows and *N* time points, as constructed in Equation (2) above. Let **Y***_i_* = **Z***_i_***F***_i_* be the PCA-reduced data matrix, where **F***_i_* is the subject’s compression matrix resulting from a subject-specific PCA with *C* < *N* columns. The temporally concatenated group data matrix is then **Y** = [**Y**_1_,**Y**_2_,…,**Y***_M_*], where *M* is the number of subjects. Lastly, a group-level PCA reduction step is applied. Let **G***^T^* = [**G**_1_*^T^*,**G**_2_*^T^*,…,**G***_M_^T^*] be the group compression matrix with *K* < *C* ∗ *M* columns, resulting from a group-level PCA applied to the group data matrix **Y**. The compressed data matrix is then as follows:(3)$$X=YG=\left[\begin{array}{ccc} Z_{1}F_{1}G_{1} & \cdots & Z_{M}F_{M}G_{M} \end{array}\right]$$

The matrix **X** will be input to both the ICA and hPCA. Note that each individual subject’s data can be back-reconstructed by multiplying by the appropriate inverses of **F***_i_* and **G***_i_*.

### 2.3. Independent Component Analysis (ICA)

Spatial ICA was performed using GIFT v4.0b, https://trendscenter.org/software/ (accessed on 26 March 2024). Although ICA shares many similarities with global PCA, differing primarily by maximizing statistical independence rather than enforcing orthogonality between components, both methods differ from the sequence of nested local PCAs in the hPCA algorithm detailed below.

For each experiment, the number of estimated components was set as the true number of SimTB components for each experiment in each simulated dataset (e.g., 39 for experiment 3L3D). While the true number of components is unknown in real data, this choice allowed ICA the best chance at accurately reconstructing the full hierarchy. Two PCA data reduction steps were then used. For the subject-level PCA, 295 components with non-zero eigenvalues were retained. For the group-level PCA, the number of ICA components to be estimated was retained. ICA was carried out using the FastICA algorithm [32], with 15 bootstrap replications in ICASSO to improve the estimated component reliability [33]. The resulting component spatial maps and time series were back-reconstructed with GICA3 and scaled to z-scores [31].

### 2.4. Hierarchical Principal Component Analysis (hPCA)

Hierarchical principal component analysis is a multiresolution analysis technique similar to wavelet analysis and hierarchical clustering analyses [26]. It is described by the pseudo-code in Algorithm 1 and illustrated graphically in Figure 1.
**Algorithm 1:** Treelets hierarchical principal components analysis1.**Input:** Data vectors {*x_v_*}*_v_*_=1,2,…,*V*_, similarity measure Sim(*x*, *x’*), *k*-dimensional principal components analysis PCA*_k_*(*x*, *x’*, …)2.𝒜 ← {*x_v_*}*_v_*_=1,2,…,*V*_                      # active set initialized with all data vectors3.𝒯 ← ∅4.**for** *v* ← 1, 2, …, *V* **do**5.    𝒯 ← 𝒯 ∪ {(*x_v_*, **0**, 0)}                      # initialize leaf nodes with datum, vector of 0 s, level 06.**end for**7.**for** *l* ← 1, 2, …, *V* − 1 **do**8.    *x_i_*, *x_j_* ← arg max Sim(*x_m_*, *x_n_*)         *x_m_*, *x_n_* ∈ 𝒜9.    (*x_s_*, *x_d_*, λ_1_, λ_2_) ← PCA_2_(*x_i_*, *x_j_*)                    # outputs PCs and eigenvalues10.    𝒜 ← 𝒜 \ {*x_i_*, *x_j_*}                         # remove from active set11.    𝒜 ← 𝒜 ∪ {*x_s_*}                          # leading PC input to higher levels12.    𝒯 ← 𝒯 ∪ {(*x_s_*, *x_d_*, *l*)}13.**end for**14.**Return:** tree 𝒯

Hierarchical PCA represents the internal structure of data as orthonormal basis functions structured as a hierarchical tree, with sum and difference variables at each level summarizing the merger of branches. Each hPCA level merges a pair of variables with a standard PCA. Each step of the hPCA algorithm generates two new variables of the same dimensions as the original data: the “sum” and “difference” variables *x_s_* and *x_d_*, respectively. These are the leading and trailing principal components of the level’s local PCA. Thus, two new components are generated at each level and this number is constant at all levels. Only one component, *x_s_*, is incorporated in subsequent levels. The maximum number of possible levels is the number of input variables minus one. For all of the above experiments, a full hierarchy was constructed with 1024 − 1 = 1023 levels. The overall sequence of mergers is structured as hierarchical decomposition T. Additionally, the sequence of merged indices creates an associated dendrogram D. Lastly, an eigenvalue pair (*λ*_1_, *λ*_2_) is associated with each merger for every level. Eigenvalues will subsequently be used for statistical inference on hPCA levels in Section 2.5 below.

The computational complexity of the hPCA algorithm itself is O(*LV*) operations, where *L* is the height of the tree and *V* is the number of variables [26]. Additional computational cost is added by calculating the similarity matrix. For simple correlations, this complexity is O(min(*VN*^2^, *NV*^2^)) operations by singular value decomposition, where *N* is the number of time points. The computational complexity and memory requirements of the algorithm can be minimized by storing the similarity matrix and tracking local changes.

The current work applies the treelets hPCA algorithm to fMRI data, using signed correlation as the similarity measure. The original variables are voxel time series, and the resulting sum variables represent the time series for large-scale networks of interconnected voxels, with each level encompassing increasing widespread activity within the brain. To facilitate group-level analyses, subject-level data were individually compressed and concatenated to form a group-level reduced data matrix, as detailed in Section 2.2 above. All voxel time series were used as input for the treelets hPCA algorithm in the current work.

The goal of the current analysis is to estimate known patterns of activity at all levels of the hierarchy, rather than find a single a priori spatial scale valid for most analyses. This multiscale approach is in contrast to a default a priori ICA model order (e.g., 20 components), or a single dendrogram cut used for hierarchical clustering analysis. To demonstrate the validity of this technique, hPCA was first applied to simulated fMRI data with a known correlation matrix with a hierarchical nested-block structure. At each level of the resulting hierarchy, the leading PC was then associated with the spatial structure in the original data through being correlated with all voxel time series. These experiments on a simulated dataset will demonstrate that hPCA can successfully capture known patterns of activity of the hierarchy.

Two fundamental problems remain with the hPCA approach. First, voxel-level fMRI data are heavily redundant, with neighboring voxels frequently differing only by a small amount of artifactual and independent noise signals. The low-level early mergers of the hPCA algorithm thus serve a denoising function, producing a series of nearly identical sum variables containing redundant information. Thus, a parametric method of identifying levels resulting in redundant mergers is highly desirable. Second, hPCA constructs a full hierarchical tree with a single root. Connectivity within the brain, however, may not be a full tree, but more likely is structured as a forest of unconnected trees, with the canonical large-scale networks at the roots. A parametric method for identifying levels merging unrelated variables is also desirable. For practical applications, parametric methods are needed due to the size of fMRI data (90,000 or more voxels).

The above two problems have previously been investigated in a non-hierarchical PCA and addressed with standard parametric statistical tests. For example, the denoising properties of PCA are a well-studied problem [34] with parametric statistical tests based on the magnitude of eigenvalues, a solution applicable for hPCA and described in detail below.

### 2.5. Statistical Inference on hPCA Levels

Following construction of the full hierarchy with hPCA, a smaller set of specific levels was selected for more detailed analysis using statistical methods. This is beneficial in practical applications, since it reduces a potentially large hierarchy to a manageable subset. For example, in experiment 5L2D, a hierarchy was constructed with 62 components (see Section 2.1). When applied to this dataset, hPCA returned 1023 levels. Of these results, only 62 levels were expected to contain relevant information. To identify these levels, two statistical tests were applied to each hPCA level individually. *p*-values for each test in each level were calculated and corrected for multiple comparisons, as detailed below. Only levels that passed both of these tests were compared to the ground-truth components, greatly decreasing the complexity of follow-up analyses.

To facilitate the applied statistical tests below, note that the hPCA algorithm is a series of PCAs, the similarities between variables are Pearson’s correlations, and the data are approximately Gaussian-distributed.

Levels merging redundant variables can be identified by testing the magnitude of the smallest eigenvalue generated at each level. Redundancy in the input variables will result in the leading PC containing nearly all of the shared variance, while non-shared variance in the trailing PC will contain a negligible amount of noise. As a consequence, the leading eigenvalue will be maximal, while the smallest eigenvalue will be near zero.

In the current work, the Smallest Eigenvalue Test (SET) [34] was applied to test the magnitude of the second eigenvalue that summarizes the amount of non-shared variance in the level’s local PCA. The null hypothesis is that the smallest eigenvalue *λ*_2_ contains a non-trivial amount of variance in the local PCA model,
(4)$$H_{0}:\lambda_{2}\geq \gamma ,$$
against the alternative that *λ*_2_ < *γ*, where *γ* > 0 is a cutoff parameter. The null hypothesis is rejected if
(5)$$\lambda_{2}<\gamma -\frac{\sqrt{2}\lambda_{2}}{\sqrt{K-1}}z_{\alpha },$$
where *K* is the number of observations (e.g., compressed and concatenated time points) and *z*_α_ is the upper significance point of a standard normal distribution N(0, 1) with significance level α. The resulting test has a significance level (i.e., *p*-value) of α. For simulated fMRI datasets with a known contrast-to-noise ratio, *γ* can be calculated exactly using Equation (2) as *γ* = 0.4 (see Appendix B). If H_0_ is rejected for a level, this indicates that the leading principal component contains most of the variance in the local PCA model and consequently can be used as a summary of the relevant information contained in the merged variables. Any differences between the merged variables are relatively small in comparison, likely resulting from artifactual noise as in Equation (2), and can be excluded from further analyses. By repeatedly applying the Smallest Eigenvalue Test to each hPCA level, a sequential series of such exclusions can be efficiently summarized using the last PCA merger in the series, greatly simplifying the analysis by focusing on non-redundant features in the data. In the current work applying hPCA to fMRI data, this simplification is expected to occur during initial levels that merge nearly identical voxels that differ only in added noise.

Levels merging unrelated variables can be tested based on their similarity value, i.e., the correlation coefficient in the current work. The null hypothesis is that the population correlation coefficient ρ ≤ 0, against the alternative ρ > 0. Using the t-approximation to the sample correlation *r* under the null hypothesis, the test
(6)$$\sqrt{K-2}\frac{r}{\sqrt{1-r^{2}}}>t_{K-2}\left(\alpha \right)$$
has significance level α, where *t_K_*_−2_(α) is the one-tailed significance point of a t-distribution with *K* − 2 degrees of freedom. A one-tailed *t*-test was applied in order to exclude negative correlations from the analysis, indicating levels merging dissimilar variables.

In both tests above, the sample size was the length of the compressed and concatenated time dimension *K*, rather than the original number of subjects. Finally, a multiple comparison procedure was applied using Bonferroni correction to control the family-wise error (FWE) rate [35]. Since there were two tests applied at each level (SET and *t*-tests) and 1023 levels in all experiments (calculated as 1024 voxels − 1), a corrected p_FWE_ = 0.001 corresponded to *p* = 4.9 × 10^−7^ (uncorrected). All statistical tests were applied following hierarchy construction, in order to select significant levels for back-reconstruction.

### 2.6. Back-Reconstruction of hPCA Component Time Series and Spatial Maps

Following statistical filtering with correction for multiple comparisons, subject-specific time series for significant group hPCA levels were estimated using GICA3 back-reconstruction [31]. This procedure reverses the initial group and subject-level data compressions by multiplying by the inverse of the respective compression matrices, resulting in subject-specific time series. Subject-specific spatial maps were then back-reconstructed using orthogonal projections similar to wavelet analyses [36]. The individual steps in this procedure are described below, including projecting group-level results into individual subject subspaces and resolving potentially correlated time series within a nested spatial hierarchy.

Let **X** = [**Z**_1_**F**_1_**G**_1_,…,**Z***_M_***F***_M_***G***_M_*] be the *V*-by-*K* compressed data matrix from Equation (3). Let † indicate the generalized matrix inverse and note that **F***_i_*^†^ = **F***_i_^T^*. Similarly, **G***_i_*^†^ is the generalized inverse of **G***_i_*.

In the ICA data model, **X** = **SA**, where **S** and **A** are the group-level source and mixing matrices, with dimensions *V*-by-*K* and *K*-by-*K*, respectively. GICA3 [31] obtains individualized source time series by back-projecting **A** onto the subject-specific subspace. For instance, time series for all components for subject *i* is back-reconstructed as **R***_i_* = **AG***_i_*^†^**F***_i_^T^*. More specifically, if **a***_k_* is the *k*th row of **A**, the time series for component *k* for subject *i* is back-reconstructed as the row vector **r***_ki_* = **a***_k_***G***_i_*^†^**F***_i_^T^*. Similarly, for hPCA, if **h** is the output from any hPCA level (e.g., **h** = *x*_s_ in Algorithm 1), then the subject-specific back-reconstructed time series **h***_i_* is
(7)$$h_{i}=hG_{i}^{†}F_{i}^{T}.$$

One way to estimate the corresponding spatial map for this component would be to correlate **h***_i_* with all other voxels for subject *i*. In the fMRI literature, this is referred to as seed-based functional connectivity analysis. A similar approach is applied below to estimate subject-specific spatial maps for hPCA. Spatial maps were then used to characterize the specialized processing encompassed by the hPCA level, based on its neuroanatomical features. Additionally, differences in spatial maps were enhanced by partially orthogonalizing the hPCA time series, similar to a partial correlation analysis.

Analyses of time series frequently apply partial correlations, in order to increase specificity and reduce any latent statistical dependencies between the time series. This is feasible for ICA, since ICA algorithms minimize the amount of spatial dependence between sources and thus their associated time series. However, this approach is not feasible for hPCA due to the extensive dependence between hierarchically nested levels. A similar orthogonalization approach is developed for hPCA below, using tree-independence in the context of the hierarchical dendrogram.

Hierarchical PCA time series were orthogonalized with respect to all time series they are independent of, with respect to the hPCA dendrogram. This is termed tree-independent orthogonalization. For example, in Figure 1, tree-dependency is illustrated with directed arrows. Each PCA time series (displayed in purple) is tree-independent of all voxel time series (displayed in red) that are not connected to it by a directed arrow. Conversely, the tree-dependencies of any individual hPCA time series in purple can be determined by tracing the arrowheads back to a subset of the voxel time series in red. Tree-independent time series are then the complement, i.e., all basis time series without arrowheads pointing to the selected hPCA level. Thus, the hPCA time series in the left middle section of the figure is tree-independent of the voxel time series located on the lower right and vice versa. In this case, tree-independent orthogonalization of this hPCA time series would regress out the latter two voxel time series. Similarly, the hPCA time series located at the top of the figure is tree-dependent on all four voxel time series. Tree-independent orthogonalization would leave this time series unchanged, since it is dependent on all time series in the basis set. Lastly, in the current work, the basis set is the level chosen using the SET to minimize computational cost.

Spatial maps were constructed using tree-independent orthogonalized time series in a seed-based functional connectivity analysis. Each hPCA time series was correlated with all voxel time series, and the resulting spatial maps were centered and scaled to z-scores. See the Appendix A for comparisons of orthogonalized and non-orthogonalized spatial maps for each experiment (Appendix A).

### 2.7. Comparison of ICA and hPCA

To facilitate visual comparison between ICA and hPCA spatial maps in the simulated datasets, all spatial maps in the figures are vectorized and displayed in rows. In Figure 3, Figure 4 and Figure 5, horizontal axes then correspond to a spatial row of voxels, while different ICA or hPCA components are arranged along the vertical axes. Since not all ICA components converged, and since the total number of significant hPCA levels may differ from the true SimTB components, the resulting displays of ICA and hPCA spatial maps may differ in size when displayed together in figures. In all figures, plotted z-score scales were standardized across experiments, including for hPCA and ICA reconstructions of the same experiment, but may differ between experiments.

Following Calhoun et al. [19] and Beckmann and Smith [18], accuracy was measured by correlating [37] the back-reconstructed ICA and hPCA spatial maps and time series with the true component spatial maps and time series from SimTB. For each true SimTB component spatial map for each subject in each experiment, the “best” match was defined as the ICA or hPCA spatial map with the highest absolute correlation coefficient. Best matches for SimTB component time series were defined with the same procedure.

The analysis intended to analyze an entire hierarchy of components with a single method, either using ICA or hPCA. Notably, since true components in the analysis may fail to be estimated in the ICA algorithm or be excluded during hPCA statistical filtering, the matching procedure was not exclusive. Thus, a single ICA or hPCA component could be the best match for multiple SimTB components, but each SimTB component can have only one best match.

The overall accuracy of each analysis type, hPCA or ICA, was compared using paired *t*-tests, with significance being defined at *p* < 0.05. The distributions of absolute correlations for each SimTB experiment and analysis type were displayed as violin plots. Using this display format, any potential differences in back-reconstruction accuracy over hierarchy levels would appear as a multimodal distribution.

### 2.8. Human Connectome Project Data and Analysis

Data used in the preparation of this work were obtained from the Human Connectome Project (HCP) database, https://ida.loni.usc.edu/login.jsp (accessed 26 March 2024). The HCP (Principal Investigators: Bruce Rosen, M.D., Ph.D., Martinos Center at Massachusetts General Hospital; Arthur W. Toga, Ph.D., University of Southern California; Van J. Weeden, M.D., Martinos Center at Massachusetts General Hospital) is supported by the National Institute of Dental and Craniofacial Research (NIDCR), the National Institute of Mental Health (NIMH), and the National Institute of Neurological Disorders and Stroke (NINDS). Collectively, the HCP is the result of efforts of co-investigators from the University of Southern California, Martinos Center for Biomedical Imaging at Massachusetts General Hospital (MGH), Washington University, and the University of Minnesota. MRI acquisition parameters have been previously published [38]. All subjects provided informed consent [39].

A subset of 500 healthy participants selected at random from the HCP S1200 release, https://www.humanconnectome.org (accessed 26 March 2024) was included in the current investigation (men = 226, women = 274, age = 28.7 ± 3.7 years). Individual resting-state fMRI scans were acquired and preprocessed by the HCP. Common noise sources were identified and removed using ICA-FIX by the HCP prior to functional connectivity preprocessing [38,40].

Data were further preprocessed for functional connectivity analyses using a pipeline validated for multiband acquisition [41]. All subjects were screened for excessive gross motion, defined as mean relative root RMS > 0.2 mm or more than 1/6 of individual volume RMS > 0.25 mm. The initial 10 s were discarded to allow for field stabilization. All volumes were spatially smoothed with kernel size FWHM = 6 mm. A nine-parameter nuisance regression model (six movement parameters, CSF, WM, and global gray matter) along with frequency filtering (0.009, 0.08 Hz) was then applied simultaneously [42]. The resulting denoised time series were extracted for all gray-matter voxels, with tissue boundaries being defined using the Hammers anatomical probability atlas [43]. As detailed above for ICA and hPCA on simulated datasets [31], subject- and group-level PCA data compression steps were applied with 300 components being retained at each. The group-level PCA was estimated with the memory-efficient Multi Power Iteration (MPOWIT) algorithm [44].

Following temporal concatenation and data compression, hPCA with statistical filtering as above was applied to all gray-matter voxels. Back-reconstructed spatial maps from significant levels were classified using correlation with template spatial maps for known Intrinsic Connectivity Networks (ICNs) [45]. Non-orthogonalized time series were used in back-reconstruction of the HCP data due to difficulties encountered in estimating higher levels of a hierarchy in simulated datasets, as detailed below. The potentially large computational demands of a voxel-level analysis were managed using memory-efficient libraries [46] and low-level programming [47].

### 2.9. Data and Code Availability

All data used in our analyses were generated using publicly available software and scripts as indicated above. Simulated datasets were created using the SimTB toolbox v18, https://trendscenter.org/software/ (accessed on 26 March 2024), extended to simulate hierarchies using custom scripts, https://github.com/koreywylie/SimTB_hierarchies (accessed on 26 March 2024). The data compression scripts, https://github.com/koreywylie/PCA.Rachakonda2016 (accessed on 26 March 2024), and hPCA scripts, https://github.com/koreywylie/HierarchicalPrinCompAnalyses (accessed on 26 March 2024), are freely available.

## 3. Results

In the following portion of the paper, Section 3.1 describes the correlation matrices for all simulated datasets. Section 3.2 then describes the outcome of the statistical filtering method applied to these datasets: specifically, how hPCA spatial maps were selected for subsequent analyses. Section 3.3 offers a detailed qualitative comparison of the spatial maps estimated for these datasets, as compared to the ground truth. Section 3.4 then quantitatively compares the accuracy of ICA and hPCA. Lastly, Section 3.5 demonstrates how the hPCA applied to real fMRI data reconstructs known hierarchical processing systems.

### 3.1. Group Correlation Matrices

A spatial hierarchy of networks was readily apparent for all three experiments in the group-averaged correlation matrices (Figure 2). In all cases, the correlation matrices showed a series of nested blocks of highly correlated voxels, with minimal correlations between unrelated blocks. Rician noise in Equation (2) is largely averaged out at the group level but is present on close inspection. The components with the smallest spatial maps showed the strongest correlations, progressively decreasing in magnitude as the block size increased. The block sizes in experiments 5L2D and 3L3D were evenly distributed. Experiment 3L3Ddirich featured a wide range of randomized block sizes, ranging from 2 to 505 voxels, with the correlation strength corresponding to positions within the hierarchy rather than block size. Consistent with the generative model for all experiments, no correlation matrix featured negative correlations.

### 3.2. Statistical Filtering of hPCA Levels

A hierarchical PCA constructs a hierarchy proportional to the size of the data, and as such, 1023 levels were estimated for all experiments in this study. Many earlier levels merge neighboring voxels, resulting in redundant time series differing only in their voxel-specific noise. In contrast, the highest levels of the hierarchy potentially combine unrelated variables, such as the final two two-way mergers that combine the largest blocks of voxels in experiments 3L3D and 3L3Ddirich. To exclude these possibilities from further analyses, parametric statistical inference methods were applied to filter the hPCA levels and limit the analysis to significant features in the hierarchy.

Levels merging redundant variables were identified by testing the magnitude of the smallest eigenvalue from each level’s local PCA using the Smallest Eigenvalue Test [34]. This statistical technique tests a level for exclusion from more detailed analyses of hPCA levels by identifying levels merging nearly identical time series in Equation (2). A series of levels flagged by the SET indicates the construction of a functionally coherent hPCA component, similar to a region of interest in fMRI. Individual voxels within this component, along with the associated lower hPCA levels, do not contain relevant information to the larger region and can be excluded from further analysis. This method was used to estimate and identify the smallest components from the lowest levels of the true hierarchy for each experiment.

The SET successfully identified a set of hPCA components that matched the set of smallest true SimTB components for all three experiments, while excluding earlier levels. Since each hPCA level decreases the number of active variables by one, the size of this set can be calculated by subtracting the index of the level flagged by the SET from the number of voxels (e.g., 1024 voxels − 992 = 32 active variables at level 992). This set of active variables from this level was then used as a summary of earlier redundant mergers.

In experiment 5L2D, the size of the set of smallest true components (2^5^ = 32) was in good agreement with the size of the level selected by the SET (Table 1; level 992 with 32 active variables, *p* < 10^−7^). Similar results were obtained for experiment 3L3D (expected 3^3^ = 27 components; selected level 997 with 27 active variables, *p* < 10^−7^) and experiment 3L3Ddirich (expected 3^3^ = 27 components; selected level 997 with 27 active variables, *p* < 10^−7^). These results support the feasibility of the SET for filtering out redundant mergers.

Levels merging unrelated variables were identified by testing the hPCA similarity measure. In the current analyses, this was the correlation coefficient between the merged variables. In experiment 5L2D, the highest three levels were identified as unrelated (Table 2, all *p* > 0.001 corrected). This number was greater than expected by two, since only the final level would have merged the two unrelated largest blocks, while all other levels would be expected to merge correlated smaller blocks (see Figure 2, left). Similar exclusions of the highest levels were also obtained in experiments 3L3D and 3L3Ddirich (Table 2, all *p* > 0.001 corrected). These results suggest that the hPCA algorithm may have difficulty estimating the highest levels of the hierarchy, or that testing the similarity of merged variables is overly conservative and may exclude these levels.

### 3.3. Evaluation of hPCA Spatial Maps

Following the statistical filtering, the remaining hPCA spatial maps for each experiment were compared to SimTB ground-truth spatial maps to determine if any hierarchy levels were not estimated.

For experiment 5L2D, the hPCA spatial maps included nearly all ground-truth spatial maps, except for the two largest blocks from the highest hierarchy level (Figure 3, bottom left). It is possible that the exclusion of these two higher-level blocks may be an artifact of statistical filtering and that information on networks may be present in their spatial maps. To investigate this possibility, time series and spatial maps were reconstructed for these levels using Equation (7). However, z-scores in the back-reconstructed spatial maps for flagged levels 1021–1023 were nearly zero for all voxels (Appendix A), validating their exclusion by statistical filtering. This suggests that hPCA encounters difficulties when estimating the highest levels of a hierarchy, corresponding to the largest and most weakly connected blocks in the group-averaged correlation matrix (see Figure 2, left). As expected, tree-independent orthogonalization during back-reconstruction sharpened the distinctions between components and levels, increasing the z-scores (Appendix A).

In experiment 3L3D, the three-way split at the top-two hierarchy levels corresponded to paired two-way mergers in the hPCA spatial maps (Figure 4). Similar to the previous experiment, hPCA spatial maps included all ground-truth spatial maps. Tree-independent orthogonalization, however, removed a large portion of the shared variance within the highest hPCA levels. The resulting spatial maps for these higher levels closely resembled the lower levels, rather than their aggregation in the true spatial maps. In this case, non-orthogonalized spatial maps more closely resembled the spatial structure of the ground-truth spatial maps for these levels with relatively weaker z-scores (Appendix A). This also supports the observation that hPCA encounters difficulties in estimating the highest levels of a hierarchy, consistent with results from the previous experiment.

In experiment 3L3Ddirich, as in 3L3D above, the three-way split at the upper hierarchy levels corresponded to paired two-way mergers in the hPCA spatial maps (Figure 5). These hierarchical PCA spatial maps included nearly all ground-truth spatial maps, including two out of three components summarizing activity from the top level of the hierarchy. Tree-independent orthogonalization during back-reconstruction removed a large portion of the shared variance within the highest hPCA levels, with the resulting highest spatial maps resembling lower levels in some cases. Non-orthogonalized spatial maps more closely resembled ground-truth spatial maps for higher levels with relatively weaker z-scores (Appendix A).

In summary, hPCA with statistical filtering was able to estimate spatial maps for nearly all levels in the spatial hierarchically structured data. This technique was robust to three-way splits and spatial heterogeneity, as observed in experiments 3L3D and 3L3Ddirich. The estimation of the lowest levels, however, was more successful than that of the highest levels in all experimental hierarchies investigated.

### 3.4. Comparison of ICA and hPCA Spatial Maps and Time Series

For all three experiments and in contrast with hPCA, ICA only estimated components from the lowest level of each hierarchy (Figure 3, Figure 4 and Figure 5). Additional ICA components appeared as unstructured noise, unrelated to ground-truth components. Furthermore, careful inspection of the ICA spatial maps revealed that the ICA estimation of components from the lowest level of the hierarchy was incomplete. In experiments 5L2D and 3L3D, ICA appeared to have incorrectly combined two independent low-level components with positive and negative weights (Figure 3 and Figure 4, bottom right), while one low-level component was missing entirely in experiment 3L3Ddirich (Figure 5, right). The components successfully estimated with ICA, however, showed higher specificity and greater z-scores when compared to their counterparts estimated with hPCA.

The observed low accuracy for ICA may depend on the model order. To investigate this possibility, a multi-model ICA was performed, where the number of components is swept across a range of values [20,23]. This analytic method has demonstrated some ability to analyze hierarchical networks in real fMRI data [23]. However, multi-model ICA did not improve upon a single-scale ICA model with the true number of components. For all three experiments, each individual model order within the multi-model ICA estimated components at a single scale with high accuracy, while components at different spatial scales were estimated with low accuracy (Appendix A). Inspection of the spatial maps suggested that the estimation of true components with low accuracy was related to missing and incorrectly combined components, similar to the problems described above for the single-scale ICA.

These results suggest that in simulated datasets with a hierarchical correlation structure, ICA may only estimate components at a single scale of the hierarchy. Additionally, these results suggest that ICA may incorrectly combine unrelated components. Notably, in hierarchical networks where only the lowest levels are of interest, these results also highlight the high specificity of ICA.

A quantitative comparison of estimation accuracy for the spatial maps and time series is shown in Figure 6. For each ground-truth component in each experiment and subject, the hPCA or ICA spatial map or time series with the highest correlating “best” match was designated as the measure of back-reconstruction accuracy. This objective procedure allowed measurement of the estimation accuracy of missing true components from higher levels of the true hierarchy. Consequentially, an ICA or hPCA component could be designated the best match of multiple true components. This was especially relevant for the ICA results, where the lowest-level components could be designated the best match for multiple higher levels of the hierarchy, as can be seen in the multimodal distributions for the ICA spatial maps’ accuracy in all three experiments (Figure 6, left).

In all three experiments, hPCA showed higher mean back-reconstruction accuracy than ICA for both spatial maps and time series. This was especially notable for time series (Figure 6, right, *p* < 0.0001 in all cases according to paired *t*-test). Additionally, hPCA spatial maps were more accurate than ICA spatial maps for experiments 5L2D (*p* < 0.0001) and 3L3Ddirich (*p* = 0.014). In experiment 3L3D, the hPCA spatial map accuracy was greater than that of ICA at the trend level (*p* = 0.10).

The multimodal distribution for the ICA spatial map accuracy was attributable to ICA’s high accuracy when estimating the lowest levels of a hierarchy and difficulty with all higher levels. Consequently, the same component from a lower level was designated the best match for all higher levels (Figure 6, left). For experiment 5L2D, this resulted in a distribution with at least four modes of decreasing accuracy. For experiment 3L3D, this resulted in a distribution with three modes. For experiment 3L3Ddirich, ICA spatial maps showed a more complicated multimodal distribution, with the highly accurate matches to the lowest levels being most prominent. In all three experiments, the accuracy of ICA back-reconstruction for these lowest levels exceeded that of hPCA for spatial maps (*p* < 0.0001 in all cases). Interestingly, and in contrast to ICA’s spatial accuracy, hPCA more accurately reconstructed the true time series for the same lower levels (*p* < 0.0001 in all cases).

In summary, hPCA was able to accurately estimate nearly all levels of the hierarchy in all three experiments on simulated fMRI data. In contrast, ICA estimated nearly all components from the lowest level with great accuracy but was unable to estimate any higher levels. Additionally, hPCA estimated all components from the lowest level of the hierarchy, while ICA estimation for the same components was incomplete.

### 3.5. Hierarchical PCA Applied to Non-Simulated Neuroimaging Data

Hierarchical PCA applied to the HCP dataset resulted in a hierarchy of networks, including the major neural processing systems. The gray-matter mask encompassed 179,120 voxels in total, allowing for 179,119 hPCA levels. Level 178,627 was chosen as the base level of the hierarchy by the SET (*p* < 0.001, corrected), indicating substantial redundancy in early levels. The mergers involving the last 12 levels were not significant based on their similarity (*p* > 0.001 corrected, in all cases) and consequently were not analyzed. In each of the figures below, the principal regions of each subnetwork, similar to hubs in graph theory, can be identified as localized cortical regions with high z-scores.

Two DMN subnetworks were present at the base level, with the spatial maps being consistent with previously reported DMN subnetworks (Figure 7) [12]. DMN-A encompassed clusters in the retrosplenial, posterior inferior parietal, and ventromedial prefrontal cortices bilaterally, as well as in the parahippocampal gyri. DMN-B encompassed clusters in the temporoparietal junction, posterior cingulate cortex, dorsomedial prefrontal, and lateral temporal cortices. Their merger, labeled DMN-AB, resulted in a combination of both networks and closely resembled the regions of the canonical DMN.

Three subnetworks of the Visual Network were present at the base level, with spatial maps consistent with known subdivisions of the visual system (Figure 8). Visual Network-A encompassed regions of the dorsal visual processing stream, including the bilateral superior occipital lobe. Other visual regions, as well as the posterior superior parietal lobe, were included to a lesser extent. Visual Network-B encompassed the ventral visual processing stream, including the fusiform gyri. Visual Network-C was centered on the primary visual cortex along the calcarine sulcus. The merger of these three subnetworks, Visual Network-ABC, encompassed the visual system in its entirety. The intermediate merger, Visual Network-AB, encompassed associative visual processing regions without extending into the primary visual cortex. This sequence of mergers followed a nested spatial hierarchy within the visual system.

The SMN was subdivided into four subnetworks at the base level (Figure 9), corresponding to subregions of the homunculi of the pre- and post-central gyri. SMN-A was centered on the ventrolateral regions bilaterally, regions dedicated to orofacial processing. SMN-B was centered on the left dorsolateral region, corresponding to somatosensation and movement of the right hand. Similarly, SMN-C was centered on the right dorsolateral region, corresponding to the left hand. Lastly, SMN-D was centered on the dorsomedial regions of the somatomotor cortex, corresponding to the feet and legs. The merger of these networks, SMN-ABCD, fully encompassed all somatomotor regions.

Three subnetworks corresponded to DAN subdivisions at the base level (Figure 10). DAN-A was centered on the left middle temporal (MT) visual area, with other DAN regions encompassed to a lesser degree. In contrast, DAN-B was centered on the right MT area. Lastly, DAN-C encompassed more superior-located regions, including inferior parietal and dorsal middle frontal gyri bilaterally. The merger of these three subnetworks, DAN-ABC, closely resembled the canonical DAN. The intermediate merger, DAN-AB, encompassed bilateral MT regions with superior DAN regions included to a lessor extend. The sequence of mergers followed a nested spatial hierarchy for dorsal attention processing, similar to that observed for visual and somatomotor processing systems.

## 4. Discussion

The current investigation applied the hPCA technique to realistic simulations of fMRI data as well as a real fMRI dataset from the HCP. Three experiments were conducted to investigate the ability of hPCA to accurately reconstruct the potentially diverse branching structure in real fMRI data. For all three experiments, hPCA successfully reconstructed true components with a wide range of spatial scales from a nested hierarchical structure. This was true for a relatively deep hierarchy with many levels (5L2D), as well as hierarchies with three-way subdivisions at each level (3L3D and 3L3Ddirch) and randomized component block sizes (3L3Ddirich).

The current work is unrelated to multiresolution parcellation atlases (e.g., [48]), although there is potential overlap with the methods used to develop many such atlases. Instead, this investigation intended to develop a novel analytic method with the ability to transcend the limitations of current methods, including parcellation atlases. In particular, hPCA is data-driven and adaptable to the unique features of any dataset, in contrast to fixed and non-data-driven parcellation atlases. Recent investigations into individual-specific connectomes in healthy subjects have demonstrated a richness of features not found in group-averaged atlases [25]. Similarly, highly variable neuroanatomical boundaries are likely present in neuropsychiatric patient populations. This pathology-associated variability would be missed when applying a group-averaged parcellation atlas, one created using only healthy subjects. In the current work, individualized subject-specific spatial maps and time series for hPCA are estimated using Equation (7), as detailed in Section 2.6 above. This allows hPCA to capture regional variability in patient subgroups and facilitates comparisons with a subgroup of healthy controls. Consequently, adaptive data-driven methods that allow for individual subject variability, such as hPCA, may reveal pathology undetected by region-of-interest or parcellation atlases created using healthy subjects.

Hierarchical PCA is a multiresolution analysis method for unordered data that constructs adaptive basis functions and a data-driven tree structure [26]. These properties are advantageous for fMRI analyses for multiple reasons. First, fMRI data are unordered, because neighboring cortical regions are not always connected, and distant regions are often connected by long white-matter tracts. Additionally, multiresolution analytical methods are needed in fMRI to capture the hierarchical spatial organization of intrinsic networks, such as the subdivisions of the DMN, Visual Network, SMN, and DAN. Furthermore, the spatial maps and time series estimated in hPCA can capture features unique to each subject, resulting from the algorithm’s construction of adaptive basis functions. Lastly, hPCA is stable in high-dimensional noisy datasets, where the number of variables greatly exceeds the number of observations, and the true underlying factors may be masked by noise [26].

In contrast, ICA could not reconstruct the data’s hierarchical structure in the simulated datasets. For all three experiments, ICA could only incompletely estimate the lowest level of each hierarchy, while missing higher levels. Since ICA is not a multiresolution method, this result was expected and demonstrates the main contrast between ICA and hPCA. Although unsurprising given the strong spatial dependence with higher levels, these results based on simulated data contrast with those observed in real fMRI data, where ICA has demonstrated some ability in multiscale analyses [20,23]. This suggests that the spatial hierarchy of intrinsic networks in real neuroimaging data features more spatial independence between levels than that featured in the simplified hierarchies in the current work. Lastly, hPCA demonstrated better accuracy in reconstructing true spatial maps and time series in the original data compared to ICA. These results demonstrate the limitations of ICA in analyzing hierarchical data and the feasibility of using hPCA for such analyses.

Hierarchical PCA combines the strengths of multiresolution wavelet and hierarchical cluster analyses. An insightful thought experiment can shed light on the differences between these methods. Wavelet analyses, unlike hPCA, require spatially or temporally ordered data, such as data ordered by time or pixel locations. Consequentially, wavelets applied to fMRI data result in spatially contiguous regions at all scales. Large distributed Intrinsic Connectivity Networks, such as the disparate subnetworks within the DMN, would not be detected using this method without rearranging voxels a priori to accommodate this single network at a single scale.

In contrast, hierarchical clustering analysis is a multiresolution analysis suitable for unordered data. Notably, in contrast with hPCA, hierarchical clustering does not construct a new latent variable at each level. Instead, in hierarchical clustering analysis, a voxel is associated with one and only one network at each scale, termed “hard clustering”. Furthermore, every encompassed voxel is assigned an equal weight, outputting network spatial maps as binary whole-brain masks. Lastly, reconstructing the time series associated with each network and subnetwork is an undefined challenge in hierarchical clustering analysis, since it only groups voxels into networks without considering how their individual time series are interrelated. In contrast to wavelet and hierarchical clustering methods, a strength of hPCA is its ability to overcome the above limitations.

The experiments conducted using simulated datasets also demonstrated the limitations of the method as well as its strengths. Hierarchical PCA encountered difficulties when reconstructing the highest levels of the hierarchy. In each experiment, the largest components resulting from the initial subdivision were incompletely estimated in the 3L3D and 3L3Ddirich experiments and were missing entirely in the 5L2D experiment. Additionally, for 3L3D and 3L3Ddirich, the hierarchically orthogonal projection method used to increase spatial map specificity removed a large portion of shared variance at the highest two levels. The resulting back-reconstructed spatial maps incompletely captured the true spatial maps and more closely resembled lower levels.

The above limitations may be overcome through future refinement of hPCA. It is possible to incorporate multivariate statistical methods such as the SET directly into the hPCA algorithm, allowing for adaptive multi-way mergers. This would potentially allow for improved denoising and more accurate estimation of higher-level networks. The full dendrogram of the HCP is too large to be visualized in a single figure. Recently, the hyperbolic disk has been suggested as a natural setting for visualizing large hierarchies [49]. This possibility will be explored in future investigations. The orthogonalization procedure could be improved through L1 or L2 normed regression, potentially improving the estimation of higher-level networks. Lastly, a complete analysis of the hierarchy of networks within the HCP dataset is beyond the scope of the current work and will be detailed in a forthcoming publication. These refinements to hPCA as well as its application to additional datasets will be the topic of future investigations, including its potential to align multimodal datasets and combine single-subject studies such as subject-level ICA.

## 5. Conclusions

In summary, the current work demonstrated the potential utility of hPCA as a data-driven hierarchical analysis method for fMRI data, using simulated datasets with realistic noise structures. A parametric statistical testing procedure was developed and applied to focus the analyses on biologically relevant components of the hierarchy. Hierarchical PCA with statistical filtering demonstrated accurate reconstruction of a spatial hierarchy of networks, including spatial maps and time series for each level. In comparison, ICA only demonstrated partial construction of components from the lowest hierarchy level, with decreased accuracy overall. Analyses of simulated datasets identified difficulty reconstructing the highest levels of the hierarchy as a limitation of hPCA. Analyses of real fMRI data from the HCP demonstrated similar results, constructing a hierarchy of known networks. These results suggest the potential of hPCA as an analytic tool for investigating the brain’s processing systems and specialized subsystems, while demonstrating its current limitations.

## Figures and Tables

**Figure 1 brainsci-14-00325-f001:**
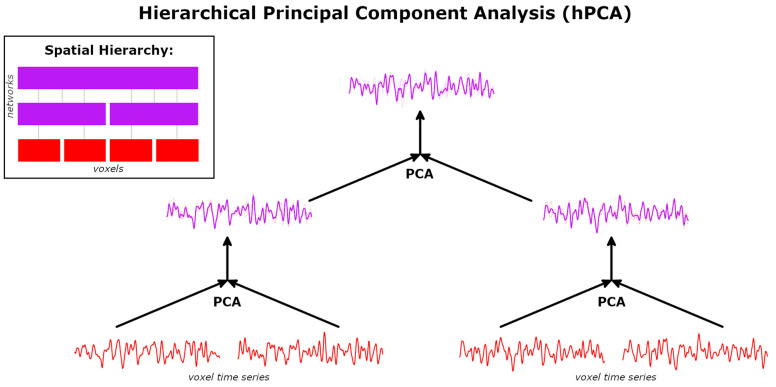
Hierarchical principal component analysis (hPCA) reconstruction of a spatial hierarchy. The hPCA algorithm is a sequence of local principal component analyses (PCA). At the initial levels, hPCA merges pairs of voxel time series (in red). At each level, a new time series is created as the first principal component (PC; solid purple line), summarizing features of both input time series (lighter dashed and dotted red lines). Inset: Example spatial hierarchy of networks of increasing scale, with encompassed voxels arranged in rows.

**Figure 2 brainsci-14-00325-f002:**
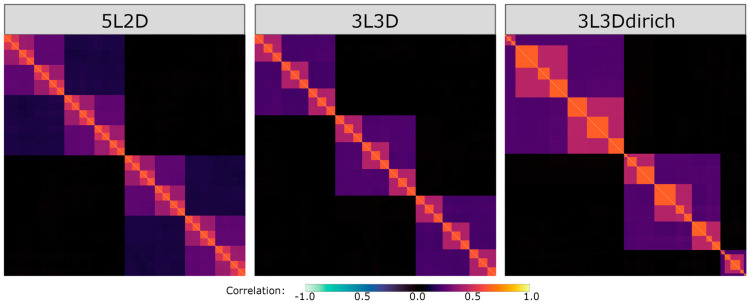
Group-averaged correlation matrices for each of the three experiments (5L2D, 3L3D, and 3L3Ddirich). Left: experiment 5L2D, a spatial hierarchy with five levels of depth and two-degree subdivision at each level. Middle: experiment 3L3D, a spatial hierarchy with three levels and three-degree subdivision at each level. Right: experiment 3L3Ddirich, a spatial hierarchy with three levels, three-degree subdivision at each level, and randomized block sizes drawn from a Dirichlet distribution with parameter alpha = (3,3,3).

**Figure 3 brainsci-14-00325-f003:**
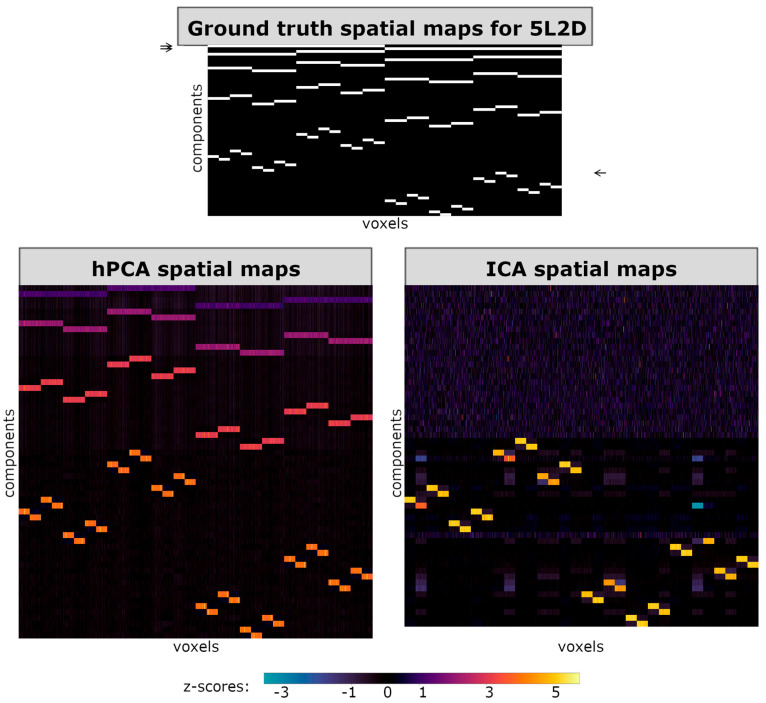
Spatial maps for experiment 5L2D. (**Top**) True group-level spatial maps used in SimTB parameters. (**Bottom left**) Mean hierarchical principal component analysis (hPCA) back-reconstructed spatial maps, following statistical filtering of levels. (**Bottom right**) Mean independent component analysis (ICA) back-reconstructed spatial maps. hPCA was able to reconstruct nearly all levels and components of the true hierarchy, with the exception of the two components in the top-most level (arrows on the top left). In contrast, ICA was only able to reconstruct most of the lowest level, with the exception of one missing component (bottom right, compared with the arrow on the top right).

**Figure 4 brainsci-14-00325-f004:**
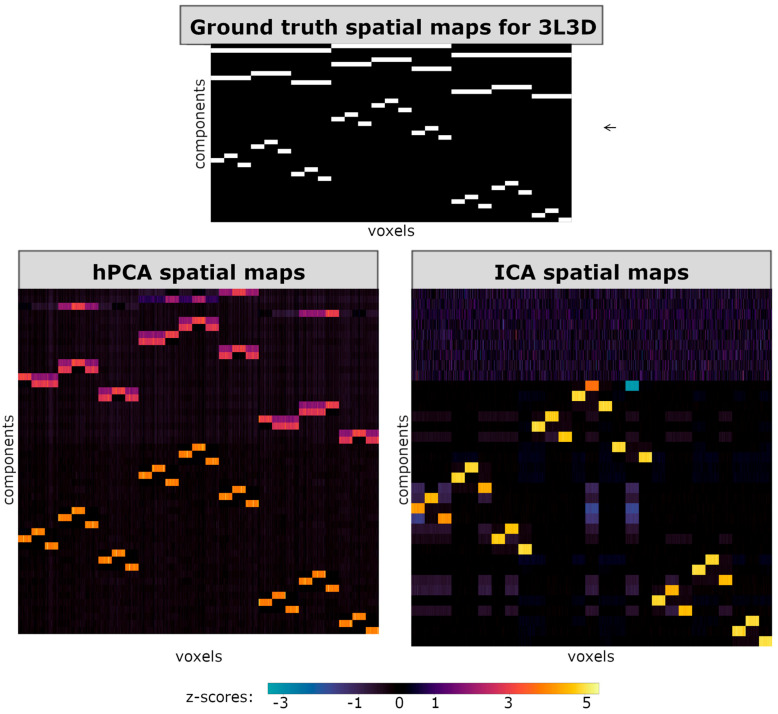
Spatial maps for experiment 3L3D. (**Top**) True group-level spatial maps used in SimTB parameters. (**Bottom left**) Mean hierarchical principal component analysis (hPCA) back-reconstructed spatial maps, following statistical filtering of levels. (**Bottom right**) Mean independent component analysis (ICA) back-reconstructed spatial maps. hPCA was able to reconstruct all levels and components of the true hierarchy. Orthogonalization during back-reconstruction, however, obscured components in the top-most levels and the resulting spatial maps resemble lower levels. In contrast, ICA was only able to reconstruct most of the lowest level, with the exception of one missing component (bottom right, compared with the arrow on the top right).

**Figure 5 brainsci-14-00325-f005:**
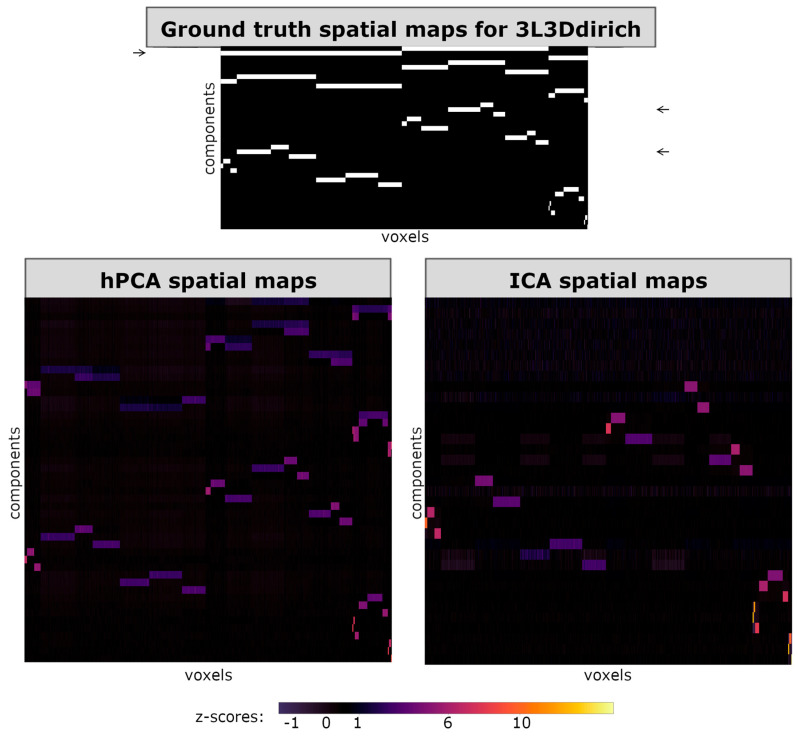
Spatial maps for experiment 3L3Ddirich. (**Top**) True group-level spatial maps used in SimTB parameters. (**Bottom left**) Mean hierarchical principal component analysis (hPCA) back-reconstructed spatial maps, following statistical filtering of levels. (**Bottom right**) Mean independent component analysis (ICA) back-reconstructed spatial maps. hPCA was able to reconstruct nearly all levels and components of the true hierarchy, with the exception of one component from the top-most level (arrow on the top left). Additionally, orthogonalization during back-reconstruction obscured components in the top-most levels and the resulting spatial maps resemble lower levels. In contrast, ICA was only able to reconstruct most of the lowest level, with the exception of two missing components (bottom right, compared with the arrows on the top right).

**Figure 6 brainsci-14-00325-f006:**
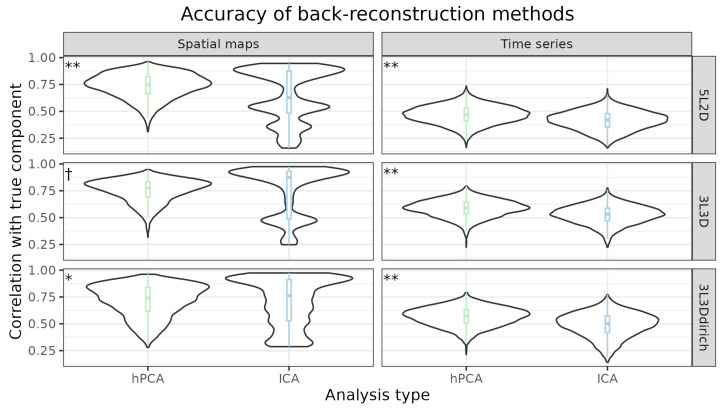
The accuracy of back-reconstruction methods. For each experiment, back-reconstructed spatial maps and time series were compared to true component maps and time series using the correlation coefficient under “best-case” matching. Resulting distributions were displayed as violin plots and box plots. For both the spatial maps and time series in experiments 5L2D and 3L3Ddirich, hierarchical principal component analysis (hPCA) was significantly more accurate than independent component analysis (ICA). For experiment 3L3D, hPCA was significantly more accurate than ICA for reconstructing time series and more accurate for spatial maps at the trend level. A multimodal spatial distribution for ICA components (left) was evident, resulting from the ICA reconstruction accuracy only being observed for the lowest level of the hierarchy; these spatial maps were chosen as the “best” match for higher levels. Legend †: *p* = 0.10, *: *p* < 0.05, **: *p* < 0.0001.

**Figure 7 brainsci-14-00325-f007:**
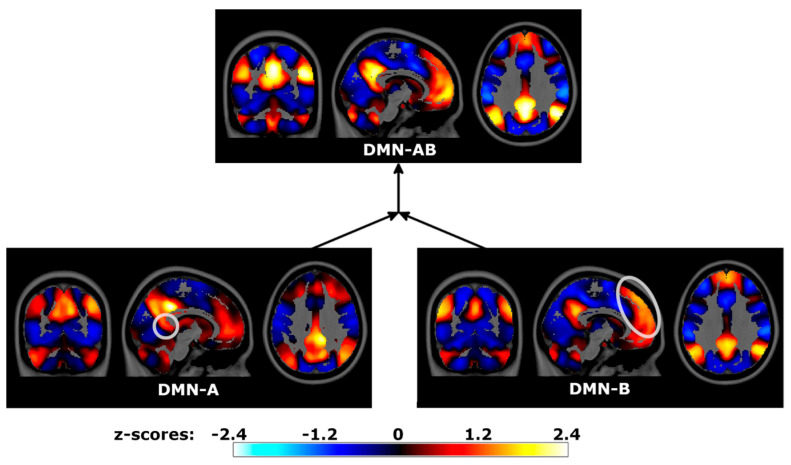
Default Mode Network (DMN) spatial maps and sequential mergers in the Human Connectome Project data. Hierarchical PCA estimated time series for two DMN subnetworks, subsequentially merged into a full DMN. Spatial maps were estimated by correlating hPCA and voxel time series, displayed in neurological coordinates. Gray circles highlight localized regional differences between merged subnetworks. DMN-A: subnetwork encompassing the retrosplenial cortex, DMN-B: subnetwork encompassing medial superior prefrontal cortices, DMN-AB: full DMN.

**Figure 8 brainsci-14-00325-f008:**
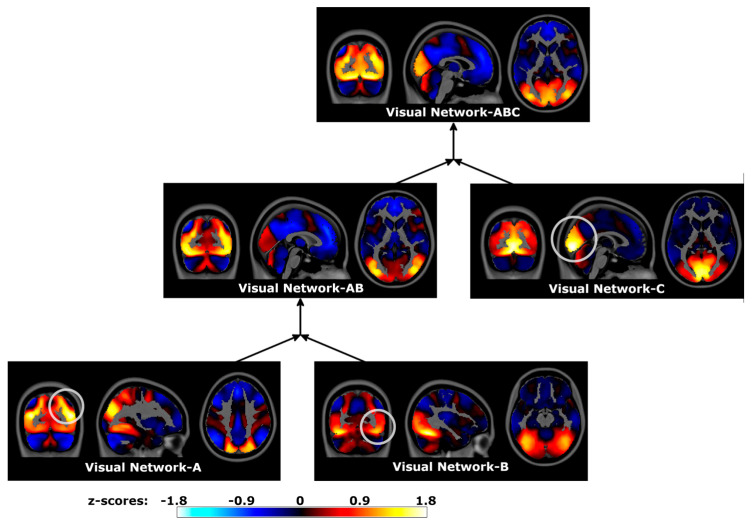
Visual Network spatial maps and sequential mergers of the Human Connectome Project data. Hierarchical PCA estimated a time series for dorsal and ventral processing streams of the visual system, as well as a time series for the primary visual cortex. Subsequent mergers then estimated time series encompassing multiple subnetworks of the Visual Network. Spatial maps were estimated by correlating hPCA and voxel time series, displayed in neurological coordinates. Gray circles highlight localized regional differences between merged subnetworks. Visual Network-A: dorsal visual processing subnetwork, Visual Network-B: ventral visual processing subnetwork, Visual Network-C: primary visual cortex, Visual Network-AB: associative visual processing regions, Visual Network-ABC: full Visual Network.

**Figure 9 brainsci-14-00325-f009:**
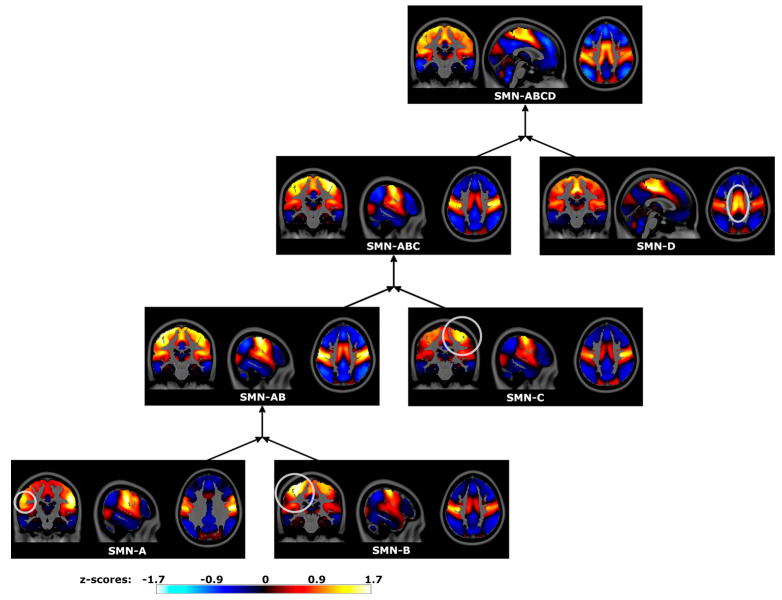
Somatomotor Network (SMN) spatial maps and sequential mergers in the Human Connectome Project data. Hierarchical PCA estimated time series for orofacial, hand, and foot regions of the somatomotor strip. Subsequent mergers then estimated time series encompassing multiple subnetworks of the Somatomotor Network. Spatial maps were estimated by correlating hPCA and voxel time series, displayed in neurological coordinates. Gray circles highlight localized regional differences between merged subnetworks. SMN-A: bilateral ventrolateral pre-central gyri (orofacial), SMN-B: left dorsolateral pre-central gyri (right hand), SMN-C: right dorsolateral pre-central gyri (left hand), SMN-D: bilateral medial somatomotor strip (foot), SMN-AB: lateral somatomotor cortices, SMN-ABC: lateral somatomotor cortices with right dorsolateral pre-central regions, SMN-ABCD: full Somatomotor Network.

**Figure 10 brainsci-14-00325-f010:**
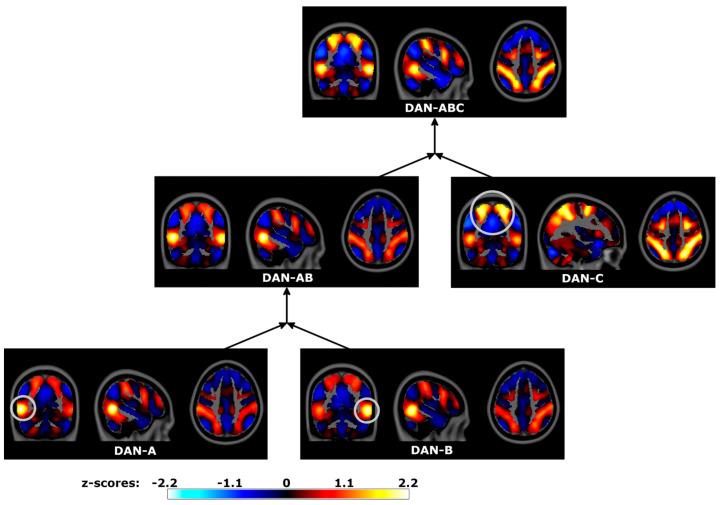
Dorsal Attention Network (DAN) spatial maps and sequential mergers in the Human Connectome Project data. Hierarchical PCA estimated a time series for each DAN subregion individually (bottom row), with sequential hPCA mergers estimating time series encompassing multiple regions (middle and top rows). Spatial maps were estimated by correlating hPCA and voxel time series, displayed in neurological coordinates. Gray circles highlight localized regional differences between merged subnetworks. DAN-A: left middle temporal (MT) visual area, DAN-B: right MT region, DAN-AB: bilateral MT regions, DAN-C: posterior superior DAN regions, DAN-ABC: full DAN.

**Table 1 brainsci-14-00325-t001:** Statistical filtering of hierarchical principal component analysis levels with the Smallest Eigenvalue Test. The Smallest Eigenvalue Test tests flagged levels merging redundant variables, with the null hypothesis being that the ratio of the second eigenvalue to the sum of both eigenvalues is non-zero. Flagged levels with p_FWE_ < 0.001 (corrected) contain redundant information and are thus excluded from further analyses. All tests corrected for multiple comparisons using the family-wise error (FWE), baseline p_FWE_ = 0.001 (corrected) = 4.9 × 10^−7^ (uncorrected).

	5L2D	3L3D	3L3Ddirich
Flagged levels	1:992	1:997	1:997
Accepted levels	993:1023	998:1023	998:1023
Max. flagged 2nd eigenvalue	0.15	0.15	0.19
Max. flagged level *p*-value	<10^−7^ *	<10^−7^ *	<10^−7^ *
Min. accepted level *p*-value	0.00000067	0.66	0.01

* *p* < 4.9 × 10^−7^ (uncorrected) < p_FWE_ = 0.001 (corrected).

**Table 2 brainsci-14-00325-t002:** Statistical filtering of hierarchical principal component analysis levels with *t*-tests. Directional *t*-tests were used to flag levels merging unrelated variables. All tests corrected for multiple comparisons using the family-wise error (FWE), baseline p_FWE_ = 0.001 (corrected) = 4.9 × 10^−7^ (uncorrected).

	5L2D	3L3D	3L3Ddirich
Flagged level	1021	1018	1018
Correlation	0.005	0.26	0.26
*p*-value	0.46	0.0000017	0.00000062
Flagged level	1022	1020	1020
Correlation	−0.001	0.26	0.18
*p*-value	0.5	0.0000017	0.001
Flagged level	1023	1022	1021
Correlation	−0.99	−0.47	0.15
*p*-value	0.99	0.99	0.005
Flagged level		1023	1022
Correlation		−0.96	−0.08
*p*-value		0.99	0.92
Flagged level			1023
Correlation			−0.79
*p*-value			0.99

## Data Availability

The data that support the findings of this study are available from the Human Connectome Project, https://www.humanconnectome.org/ (accessed on 26 March 2024). These data are publicly available to researchers who agree to the data use terms.

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
