# Peer review of "Hierarchical Principal Components for Data-Driven Multiresolution fMRI Analyses"

_brainsci, 2024, doi:10.3390/brainsci14040325_

Round 1

Reviewer 1 Report

Comments and Suggestions for Authors

The article is interesting and introduces the importance of analysis in a broad way, even a little redundant and repetitive, but the introduction lacked a bit of coverage of the literary basis on the accuracy of methods, since the proposal is methodological.

The methodology could be a little clearer, with a brief explanation of all the comparisons and the purposes for achieving the objectives.

In the results section of the article, the only observation is that the first table has a typing error, instead of the asterisk the number 1 was placed, the rest is very well developed and analyzed. In the discussion, the authors could include some other methodological methods already tested for this objective and the similarities or differences of other studies compared to what is proposed in this one.

Comments on the Quality of English Language

I have made a revision to make the text clearer and remove some typing errors.

Author Response

Point by point response in attached file

Reviewer 2 Report

Comments and Suggestions for Authors

Introduction: I read the instruction with interest. However, l found a series of conceptual and theoretical gaps that should be addressed to guide the reader and introduce the aims.

The authors addressed the question mainly using the anatomical systems in the brain. Indeed, they introduced the visual system and its functional specialization. Similarly they introduced the somatomotor system based on the topography of the homunculus. This is partially uncorrected , since several studies highlighted in the past 10-15 years the cons of this theory, overall for the motor functions. Please, revise this part and introduce new refs. Similarly, the link between somatosensory system and DMN Is not clear. DMN, as It Is widely known, is composed by several subsystems involved in different types of functions. Together these subsystems allow the DMN in its principal functions (also several) and in its crosstalk with other networks (i.e. DAN). In this way, the authors need to clarify terms such as "subnetworks"or "regions". Despite the subsystems, each network shows principal regions or hubs, which are also matter of discussion.

The spatial hierarchy is only one of the possible organizing principles in the brain. This can be relevant in part for the fMRI, but however, the authors need to be more specific and add more literature about.

As the authors' principal objective is about PCA, they highlighted the role played by ICA. However, ICA can be combined with other types of FC analysis techniques to allow at disentangling the complexity of a network or , for example , the type of level of association between 2 or more networks. However, the link between PCA and ICA is not clear. The authors could address the specific difference between independent and principal component analyses. 

Line 78. I agree with the authors about the limitations of the ICA in parcellation at a smaller scale. However, the authors, here, should need to add previous studies that tried to give a possible solution to this issue. Similarly, the authors underlined the relevance of single subject studies and the variability raised. In this case, I advise the authors to link this important concept to the ICA and FC studies and related limitations.

The title of the figure 1 should be correct adding "Figure 1. Etc". 

The authors introduced the hPCA. However, the PCA has been previously applied in studies that used different neuroimaging techniques. 

Similarly, the similarities between hPCA and clustering is not clear.

In the methods, the authors need to introduce the simulated and open access datasets with a brief description of the characteristics. After that, authors can explain the procedure, adding a figure or a flowchart. I also advise to rewrite this section with a different structure. This is crucial for reproducibility.

Similar, the authors can follow a simple checklist that can be helpful in organising these sections following point by point the items. 

Reading the methods was a bit difficult and I advise the authors to organise better this section. Indeed, a better organisation can be helpful to read and eventually replicate the method with a different dataset.gor example the authors could add different sections. Moreover, I advise to avoid to mention specific limitations in the methods, adding them in a specific section or in the discussion. 

I also advise to organise the figures and the results following a more organised structure.

Author Response

(The authors gave the same response as above.)

Reviewer 3 Report

Comments and Suggestions for Authors

My review of this manuscript is limited by the fact that I am a neuroscientist, not a mathematician.  I consider that am equipped to evaluate the rationale for application of hPCA  to investigate the organization of brain networks based on fMRI data, and to comments on the apparent utility/validity of the results they report. However, I have only a broad conceptual understanding of the technicalities of the hPCA procedure and I cannot comment on the soundness of the details of mathematical procedures employed.

The proposal that the brain networks are organized in a hierarchical manner is plausible.  There is abundant evidence in of functional subunits with the major resting state networks. It would be expects that during specific task, subunits from different networks would interact directly (eg during a simple hand movement guided by a simple static visual stimulus at fixation, it would be expected that low level visual units would interact with the hand area of the primary motor cortex). However, the large scale networks observed during the resting state indicate that the strongest interactions of specialised  sub-units during the resting state are with sub-units in the  same large scale sensory,  motor or attentional system.   It therefore makes sense to employ a technique such as hierarchical PCA to investigate the organization of networks in the resting state.   

They compared the performance of hPCA with a widely accepted ICA approach (Calhoun et al) in identifying the network structure in simulated data created in a manner that imposed hierarchical structure. Not surprisingly, overall, hPCA outperformed ICA. Nonetheless, as the authors themselves acknowledge, despite superior performance compared with ICA, their hPCA analyses of the simulated datasets had only limited success in constructing the highest levels of the hierarchy.

Their findings from the analysis of real resting state fMRI data from the Human Connectome Project are consistent with the proposed hierarchical organization.  

Thus, despite limitations of the current hPCA procedure, the approach is potentially a useful step towards a meaningful analysis of human brain networks capable of delineating functional subunits within the currently recognised resting state networks.

Author Response

(The authors gave the same response as above.)

Reviewer 4 Report

Comments and Suggestions for Authors

1- The number of components at each level does not follow a pattern in the experiments. For example, 5L2D differs from 3L3D. How do you choose the number of components for each experiment?

2- Generally, having many main components, the majority no longer express data variability. How do you know that the information generated in most of the components is important?

3- Is the number of components generated at each hPCA level the initial number plus x_d and x_s, or can it vary? How is the number of levels defined?

4- It was not clear how G^T (group compression matrix) is generated.

5- Better describe Statistical Filtering of hPCA Levels, it was not clear how it is carried out. Sometimes the explanation of the Smallest Eigenvalue Test can help with understanding.

6- In Figure 1, it is clear that the number of components generated at each level may be different from the initial one.

7- The keywords ICA and independent component analysis became redundant. Add hPCA as a keyword.

Author Response

(The authors gave the same response as above.)

Reviewer 5 Report

Comments and Suggestions for Authors

The authors of the article “Hierarchical Principal Components for Data-driven Multiresolution fMRI Analyses” propose applying the hPCA to accurately estimate spatial maps and time series from networks with diverse hierarchical structures.

The article is well-structured and properly formalized.

There are two commentaries :

1.- Figures 1 and 3 are fundamental in this work. Matrix A contains data vectors, and T creates the nodes.

In Fig 1, the voxels would be the data that is loaded into the tree. However, it is not explained whether all the voxels of the fmri are loaded or only certain nested regions.

  How is the hierarchy that is being talked about fulfilled? Or under what criteria was the hierarchy or partition generated?

2.- Equation 2 does not coincide exactly with what is reported in reference 23. Matrix A in reference 23 is the data matrix without noise. However, equation 2, between lines 152-152, considers the term epsilon 1. Why analyze it like this?

Author Response

(The authors gave the same response as above.)

Round 2

Reviewer 2 Report

Comments and Suggestions for Authors

The authors addressed my concerns. Indeed, the manuscript is interesting.

Reviewer 5 Report

Comments and Suggestions for Authors

The authors of the article “Hierarchical Principal Components for Data-driven Multiresolution fMRI Analyses” answered the observations made and modified the text in the article, which makes it clearer. For this reason, I suggest that the article be accepted for publication.